# Response of Phytoplankton Community Structure to Vegetation Restoration after Removal of Purse Seine in Shengjin Lake

Xudong Zheng, Jingwen Chen, Wenli Guo, Zhongze Zhou and Xiaoxin Ye *

School of Resources and Environmental Engineering, Anhui University, Hefei 230601, China; x19301057@stu.ahu.edu.cn (X.Z.); x19201018@stu.ahu.edu.cn (J.C.); x19301100@stu.ahu.edu.cn (W.G.); zhzz@ahu.edu.cn (Z.Z.)
* Correspondence: yexx@ahu.edu.cn

**Abstract:** Aquatic vegetation has been restored since the removal of seine nets from the lake surface of Shengjin Lake in 2018. Through four seasons of phytoplankton sampling surveys from 2019–2020, we analyzed spatial and temporal changes in phytoplankton communities, water quality, and aquatic plant recovery in conjunction with previous research literature to reveal the response mechanisms of phytoplankton community structure to rapidly recovering aquatic vegetation. The results showed that the Secchi depth increased (0.4 m to 0.7 m), the concentration of total phosphorus decreased (0.053 mg/L to 0.41 mg/L), the species of aquatic plants (5 species to 16 species), phytoplankton species (210 species to 254 species) and cell density increased after the removal of the seine. Since the removal of the seine of Shengjin Lake, the aquatic vegetation cover has exceeded 80%, the phytoplankton biodiversity has increased, and the water quality has recovered to II-III water status. Our results show that aquatic plants improve water quality through direct and indirect effects and influence phytoplankton community structure together with the water environment, which can provide guidance for the restoration situation of the middle and lower reaches of the Yangtze River through-river lake ecosystems.

**Keywords:** vegetation restoration; phytoplankton; Shengjin Lake; environmental factors

## 1. Introduction

Aquatic ecosystems are responsible for the energy flow and circulation in the food chain of lakes, and phytoplankton is the primary producer in the aquatic ecosystem (Plus et al., 2015). Because of the high sensitivity of phytoplankton to the environment, it is used as an indicator in ecological evaluation [1,2]. By studying the changes and characteristics of phytoplankton community structure, the water quality of lakes was analyzed, and the nutritional status of lakes was evaluated. Generally speaking, the change of phytoplankton community structure is mainly affected by water temperature (WT), pH, Secchi depth (SD), dissolved oxygen (DO), and other hydrological conditions. The grazing of fish and zooplankton in water is also one of the factors limiting the growth of phytoplankton [3–5]. Studies have shown that excessive inflow of nutrients such as nitrogen and phosphorus will lead to a sharp increase in the number of phytoplankton, especially Cyanophyta, which is also the cause of eutrophication in most lakes in China [6,7]. However, a single environmental factor can not effectively describe the complex dynamic changes of phytoplankton, and each environmental factor has a coordinated effect on the phytoplankton community. In order to analyze which environmental factors play a leading role in affecting the phytoplankton community structure, we conducted an in-depth study of the lake.

There is a large area of lakes in the middle and lower reaches of the Yangtze River in Anhui Province. These lakes are connected with the Yangtze River, forming the important and unique aquatic ecosystem of a river-connected lake. At the same time, these lakes

constitute both an ecological barrier and an important ecological function of the economically developed area of the Yangtze River Delta. The unique hydrological conditions formed by the connection between river-connected lake and the Yangtze River cause the lake water level (WL) to change seasonally with the Yangtze River. The changing WL, velocity, and WT are the basic elements to maintain the vitality of the wetland ecosystem, which is also conducive to the feeding and reproduction of waterfowl and fish, and improve their biological species and density [8]. During the summer, rainfall is heavy. As a result, the Yangtze River water flow increases to the lakes, leading to a rise in WL. The WL reaches its highest point in August and remains fairly constant until after September. This creates a period of high water. After September, the amount of water in the Yangtze River decreases, the lake sluice opens, and the water flows back into the Yangtze River, replenishing it. The WL drops to its lowest point in the winter (after December), and the shoals around the lake are exposed. This is the low water period. At the beginning of April, the WL gradually returns to normal [9]. Shengjin Lake is a typical river-connected lake in the middle and lower reaches of the Yangtze River. It is a National Nature Reserve in Anhui Province that protects wintering migratory birds, and more than a million waterfowls gather to overwinter every year [10]. This is the most important wintering place for whiteheaded cranes in China, an important resting place for waterbirds on the migratory route from East Asia to Australasia, and a place for fish migration and breeding in the Yangtze River [11–13]. However, starting in 1997, the local residents developed a purse seine fishery based on river crab culture in order to meet the needs of the economy. This overexploited the natural resources. Consequently, there are a series of environmental problems in Shengjin Lake: the aquatic vegetation is seriously degraded, the submerged vegetation has almost disappeared, the ecological function is degraded, the water quality deteriorates, and the eutrophication has intensified [14]; the number of birds feeding on the roots of submerged plants has decreased due to the decrease of food; purse seine blocks the hydrological connectivity between Shengjin Lake and the Yangtze River, resulting in the loss of spawning and migration channels for migratory fish, the destruction of fish physiological habits and the reduction of fish species.

The Chinese government attaches great importance to the ecological restoration of Shengjin Lake, Huayang Lake, Caizi Lake, and other lakes in the Yangtze River Basin. Since 2016, the government has implemented the Yangtze River protection policy. The government has invested hundreds of millions of funds to dismantle the seine in the lake, ban fishing and compensate the fishermen, enforcing the "no seine, no fishing, no boat" of the lake, to restore the ecological diversity of the lake and maintain the stability of wetland ecosystem function. After the removal of purse seine in 2018–2019, the species of overwintering waterbirds increased to 29, and the diversity of fish and birds increased [15]. In the investigation of Huayang Lake, the diversity of fish showed similar changes [16]. Since the complete removal of the seine net on Shengjin Lake in 2018, aquatic vegetation has recovered rapidly. Through regular monitoring of the aquatic vegetation of Shengjin Lake, we have found that the emerged plant *Zizania caduuciflora* and the floating plant *Trapa incisa* has a large distribution in the lake area of Shengjin Lake as pioneer species, especially in some waters of the upper lake (UL), where the cover of *Z. caduuciflora* and *T. incisa* is as high as more than 90%, and the submerged plant *Vallisineria spiralis* and *Myriophyllum verticillatum* are the main companion species.

Studies have shown that aquatic vegetation will either directly or indirectly affect the phytoplankton community structure, depending on the type of vegetation. Submerged macrophytes absorb a lot of nitrogen, phosphorus, and inorganic carbon through their rhizomes, their leaves block light, and their bodies secrete allelochemicals; large areas of aquatic vegetation weaken the effect of water flow and wind and impacts the hydrological conditions such as WT and DO [17,18]. On the other hand, aquatic vegetation provides a living environment for zooplankton to limit predation by large carnivorous fish, which increases the biomass of zooplankton and limits the reproduction of phytoplankton [19,20]. Therefore, in order to study the effects of aquatic vegetation restoration and lake envi-

ronmental factors on the phytoplankton community structure, long-term monitoring of Shengjin Lake is needed.

In this study, our research team regularly monitored the water environmental factors, aquatic vegetation distribution, and phytoplankton community structure of Shengjin Lake from April 2019 to January 2020 with the following purpose: (1) to find out the distribution of aquatic vegetation through field investigation and remote sensing monitoring data; (2) by analyzing the temporal and spatial changes of phytoplankton community structure and water quality data, combined with the previous research literature, the response mechanism of phytoplankton community structure to rapidly recovering aquatic vegetation was revealed. The results can provide a more comprehensive scientific basis for the management and protection of freshwater lakes in the middle and lower reaches of the Yangtze River.

## 2. Materials and Methods

### 2.1. Description of Study Area

The study area is Shengjin Lake National Nature Reserve (E116°55′~117°15′, N30°15′~30°30′), which is located in the South Bank of the middle and lower reaches of the Yangtze River in Anhui Province, China. It is connected with the Yangtze River through a waterway, Huangpeng Sluice. The main flood season of the Yangtze River is from July to September, which is basically the same as the rainy season due to the influence of rainfall in the basin. Therefore, the WL of Shengjin Lake rises and falls with the WL of the Yangtze River. The water source comes from Zhangxi River in the southeast and Tangtian River in the northeast. The lake area is 132.8 square kilometers. The WL ranges from 0.20 to 8.74 m, and the average WL is 5.21 m. The area has a subtropical monsoon climate with four distinct seasons, abundant rainfall, and sunshine. The annual average temperature was 16.1 °C, the average summer temperature was 28.8 °C, and the average winter temperature was 3.9 °C. Under the influence of the monsoon, the rainfall has obvious seasonal changes, with the largest rainfall in summer and an average annual rainfall of 1600 mm with abundant surface runoff [21].

### 2.2. Data Collection

We collected the samples upper (UL), middle, and lower (ML) of Shengjin Lake, with 8 sections and 24 sampling points (Figure 1). The sampling point SJ1~12 was located in the UL and SJ13~24 in the ML, including SJ1~2, SJ5~9 and SJ11~14, which was covered by the floating plant *T. incisa*, and SJ3, which was covered by the emerged plant *Z. caduuciflora*. From April 2019 to January 2020, we collected 4 samples that represented the water quality of the lake in spring (April), summer (August), autumn (October), and winter (January of the following year).

We used the SX751 portable water quality analyzer (Sanxin Instrument Co., Ltd. in Shanghai, China) to measure the water quality of Shengjin Lake on site, including WT, DO, Cond, and pH. We used a Hach 2100Q portable turbidimeter to calculate the turbidity (Turb). SD was measured with the Secchi disk, and WL was measured with a scale lead hammer. Chl a and nutrient concentrations were measured in the laboratory [22], and total phosphorus (TP) was determined by ammonium molybdate spectrophotometry, total nitrogen (TN) by alkaline potassium persulfate ablation, and Chl a by acetone extraction spectrophotometry under an ultraviolet spectrophotometer (UV 2450), respectively [23].

For quantitative analysis of phytoplankton, we collected 1 liter (L) of water sample and added 10 milliliter (mL) of 1% Lugol iodine solution for fixation. After letting it stand for 48 h, we slowly siphoned the supernatant to 30 mL in order to identify the phytoplankton [24,25]. Under the optical microscope, we counted 0.1 mL samples with $40 \times 10$ magnification and randomly selected 100 fields for the identification and counting of each sample [26]. We identified phytoplankton species based on morphology [27]. The biomass of phytoplankton is mainly measured by individual traits. Phytoplankton abundance and biomass were expressed as cell/L and mg/L, respectively [28].

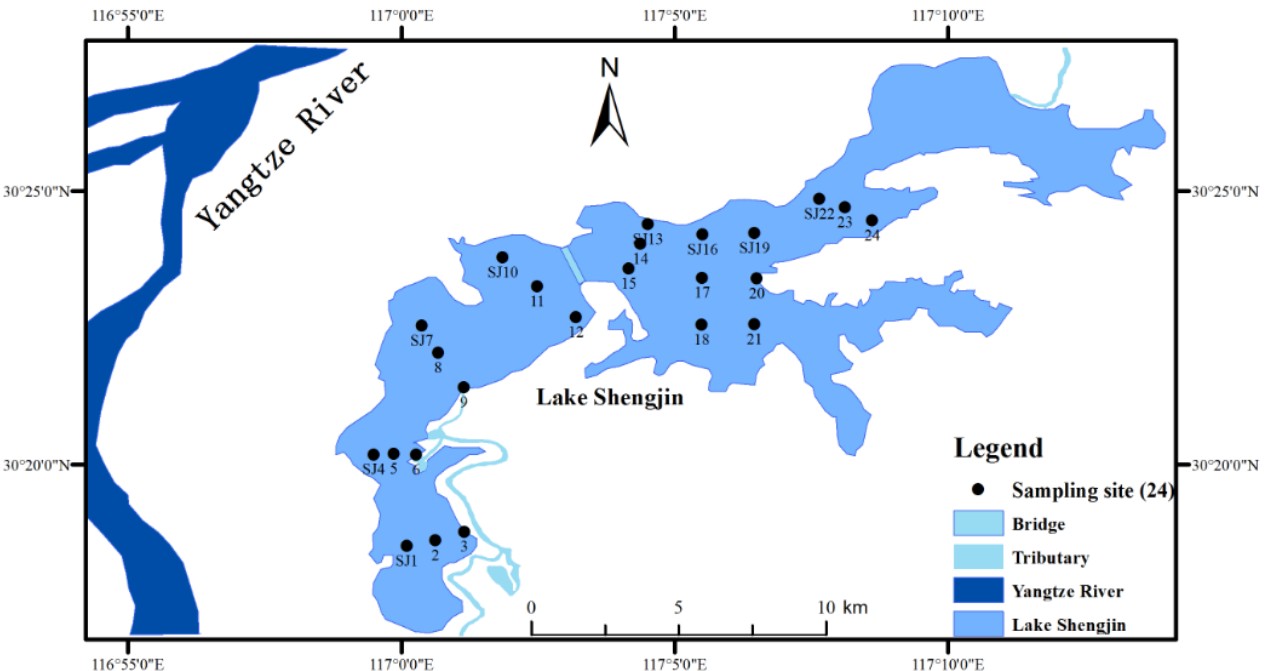

**Figure 1.** Distribution of sampling points in the study area of Shengjin Lake.

*2.3. Data Analysis*

The calculation formula of Mcnaughton dominance index (*Y*) is as follows:

$$Y = n_i/N \times f_i \tag{1}$$

In order to analyze the community structure of the phytoplankton, we applied the Shannon-Wiener index (*H′*), the Pielou index (*J′*), and the Margalef index (*D*). The calculation formulas are as follows [29]:

$$H' = -\sum_{i=1}^{s} \frac{n_i}{N} \ln \frac{n_i}{N} \tag{2}$$

$$J' = H' / \ln S \tag{3}$$

$$D = (S-1)/\ln N \tag{4}$$

where $n_i$ is the number of individuals of species *i*, *N* is the total number of individuals of all species, *S* is the total number of phytoplankton species, and $f_i$ is the frequency of individuals in the *i* species; following [30], we consider (*Y*) $\geq$ 0.02 as an indication of phytoplankton dominance.

Using IBM SPSS Statistics (Version 26.0), we conducted a one-way ANOVA method to test for significant differences between environmental parameters and phytoplankton communities' cell density between seasons. We evaluated the effect of water quality parameters on the phytoplankton density with the Pearson correlation analysis, using IBM SPSS26.0 Statistics, according to the correlation coefficient, the significant correlation between water quality parameters, and phytoplankton density was tested. We performed RDA utilizing Canoco 5 to explain the relationship between dominant phytoplankton communities and environmental factors. To reduce the effect of rare phytoplankton species, we transformed the phytoplankton data into log10(x + 1) format before data analysis. The results of RDA were visualized in the form of ordination plots based on AX 1 and AX 2 axes generated by Canoco for Windows 5 software. Environmental variables and species are designated by arrows. The position of each species in the ordination plot indicates the point corresponding to the optimum value of that species in the gradient. We produced the bar graphs

with Microsoft Excel and the spatial interpolation of cell abundance and the distribution of aquatic vegetation with the ArcGIS 10.2 software applying the kriging method.

## 3. Results

### 3.1. Environmental Characteristics Parameters

The following table shows the main environmental factors of Shengjin Lake from 2019 to 2020 (Table 1). The results revealed that there were significant differences in the seasonal and spatial distribution of WT in Shengjin Lake ($p < 0.01$). During the study period, when the WT was 8~35.7 °C, the highest WT appeared in the summer and the lowest in the winter. The pH was between 7.47~9.26, the mean was 8.68, and the whole lake was weakly alkaline. DO is significantly higher in the winter than in the other seasons, with an average of 11.75 mg/L, and the lowest in summer, with an average of 7.69 mg/L. One-way ANOVA showed that WL and SD in summer were significantly higher than those in other seasons, ranging from 0.31 m to 6.30 m and 0.07 m to 2.45 m, respectively. Pearson correlation results there was a significant correlation between WL and SD (* $p < 0.05$). The concentration of nitrogen and phosphorus is one of the important indexes to evaluate the health and self-purification of lake water. The concentration of TN ranges from 0.08 mg/L to 1.62 mg/L. The concentration of TP ranged from 0.003 mg/L to 0.158 mg/L, and the concentration of nitrogen and phosphorus in the UL was lower than that in the ML. The concentration of nitrate nitrogen ($NO_3$-N) ranged from 0.001 mg/L to 3.41 mg/L. The average concentration of $NO_3$-N in the whole lake was the lowest in the summer when the plants were growing vigorously, and it gradually increased with the seasonal changes. The average concentration of $NO_3$-N in the spring was the highest. In general, the water quality of Shengjin Lake was in the II-III standard during the study period.

### 3.2. Distribution of Aquatic Vegetation

Shengjin Lake has abundant precipitation in July and August, and the lake is in a period of abundant water. A total of 16 aquatic plants were monitored during the study. The aquatic plants detected in the spring included *Ceratophyllum demersum, Hydrilla verticillata, Utricularia aurea, T. incisa, Euryale ferox, Nelumbo nucifera, Z. caduuciflora*, etc. Figure 2 illustrates the distribution of dominant associations of main aquatic plants' spread in the lake during the summer: (1) the aquatic plant communities in the lake at SJ1~2 of the UL District were Assoziation. *Trapa incisa-N. nucifera*, associated species are *Z. caduuciflora* and *V. spiralis*, and the coverage was 95%; (2) the aquatic plant community distributed on the lake surface at SJ3 is Ass. *Zizania caduuciflora*, the coverage was 90%, associated plants are *H. verticillata, C. demersum*; (3) SJ5~9 is the central water area of the UL, and the distribution of floating plant is Ass.*Trapa* sp., and the coverage was 55%, the associated species are the emerged plant *Z. caduuciflora* and the submerged plant *M. verticillatum, H. verticillata*; (4) Ass. *Polygonum lapathifolium* is an emerged plant distributed in the water area of SJ7, and the coverage was 15%; (5) there is a large area of Ass.*Trapa incisa* near the bridge, covering more than 90%. There is no aquatic vegetation with large coverage in the open water area of the ML and a small number of Ass.*Trapa incisa* is distributed near the shore. Compared with the UL, the coverage of aquatic vegetation in the ML is greatly reduced. Compared with the spring, vegetation coverage in the summer increased greatly. As the dominant aquatic plants in Shengjin Lake are *Z. caduuciflora* and *T. incisa*, both of which are thermophilic plants. The suitable temperature for growth is 10–25 °C, and they will die in autumn and winter. On the contrary, *Potamogeton crispus* is a typical submerged plant that germinates in autumn and grows over the winter. Its buds germinate in early October. In the autumn and winter, we monitor the newly grown submerged plants *Potamogeton crispu*s. They were scattered in the shallow coastal waters while the other aquatic plants gradually disappeared.

**Table 1.** The average value of environmental parameters of Shengjin Lake.

| Environmental Factors | Spring (April) | | Summer (July) | | Autumn (October) | | Winter (November) | | Seasonal *p*-Value | Spatial *p*-Value |
|---|---|---|---|---|---|---|---|---|---|---|
| | UL | ML | UL | ML | UL | ML | UL | ML | | |
| WT (°C) | 27.66 ± 0.73 | 27.25 ± 0.31 | 34.92 ± 0.48 | 34.43 ± 0.15 | 17.35 ± 2.65 | 16.27 ± 0.26 | 8.57 ± 0.25 | 8.13 ± 0.11 | ** | * |
| pH | 8.47 ± 0.45 | 9.03 ± 0.15 | 8.6 ± 0.18 | 8.77 ± 0.1 | 8.64 ± 0.36 | 8.43 ± 0.13 | 8.77 ± 0.04 | 8.52 ± 0.18 | - | - |
| DO (mg/L) | 7.98 ± 1.02 | 8.39 ± 0.29 | 7.67 ± 0.39 | 7.72 ± 0.16 | 9.73 ± 0.17 | 9.45 ± 0.08 | 11.75 ± 0.11 | 11.77 ± 0.15 | * | * |
| Cond (Us/cm) | 116.05 ± 6.82 | 111.21 ± 2.16 | 190.57 ± 6.5 | 177.04 ± 3.36 | 247.5 ± 8.5 | 234.5 ± 7.76 | 157.73 ± 2.44 | 154.38 ± 3.52 | * | - |
| SD (m) | 1.08 ± 0.1 | 0.86 ± 0.12 | 1.37 ± 0.56 | 1.14 ± 0.19 | 0.20 ± 0.06 | 0.19 ± 0.06 | 0.16 ± 0.01 | 0.13 ± 0.02 | ** | * |
| WL (m) | 2.54 ± 0.45 | 3.66 ± 0.38 | 4.19 ± 0.30 | 5.57 ± 0.62 | 0.45 ± 0.25 | 0.97 ± 0.27 | 1.08 ± 0.23 | 1.43 ± 0.13 | ** | * |
| Turb | 4.06 ± 1.63 | 8.06 ± 1.08 | 5.54 ± 1.95 | 5.48 ± 1.68 | 41 ± 24.2 | 56.33 ± 44 | 203.72 ± 77 | 232 ± 89.37 | ** | * |
| TP (mg/L) | 0.009 ± 0.1 | 0.013 ± 0.01 | 0.027 ± 0.01 | 0.036 ± 0.02 | 0.072 ± 0.06 | 0.077 ± 0.04 | 0.051 ± 0.01 | 0.073 ± 0.09 | ** | * |
| TN (mg/L) | 0.97 ± 0.21 | 1.15 ± 0.36 | 0.57 ± 0.52 | 0.92 ± 0.35 | 1.07 ± 0.14 | 1.33 ± 0.82 | 1.12 ± 0.51 | 1.18 ± 1.08 | * | * |
| NO$_3$-N (mg/L) | 2.12 ± 0.99 | 0.82 ± 0.33 | 0.15 ± 0.05 | 0.06 ± 0.04 | 0.013 ± 0.01 | 0.14 ± 0.17 | 0.18 ± 0.02 | 0.21 ± 0.06 | - | - |
| NH$_3$-N (mg/L) | 0.09 ± 0.05 | 0.13 ± 0.1 | 0.09 ± 0.05 | 0.13 ± 0.09 | 0.89 ± 0.36 | 0.81 ± 0.23 | 0.24 ± 0.07 | 0.28 ± 0.14 | * | - |
| Chl a (µg/L) | 2.47 ± 4.72 | 3.42 ± 3.15 | 1.32 ± 0.71 | 2.58 ± 1.05 | 1.52 ± 1.1 | 0.87 ± 0.41 | 3.08 ± 0.78 | 2.92 ± 0.71 | * | * |

* $p < 0.05$ indicates significant difference at 0.05 level, ** $p < 0.01$ indicates a highly significant difference at the 0.01 level.

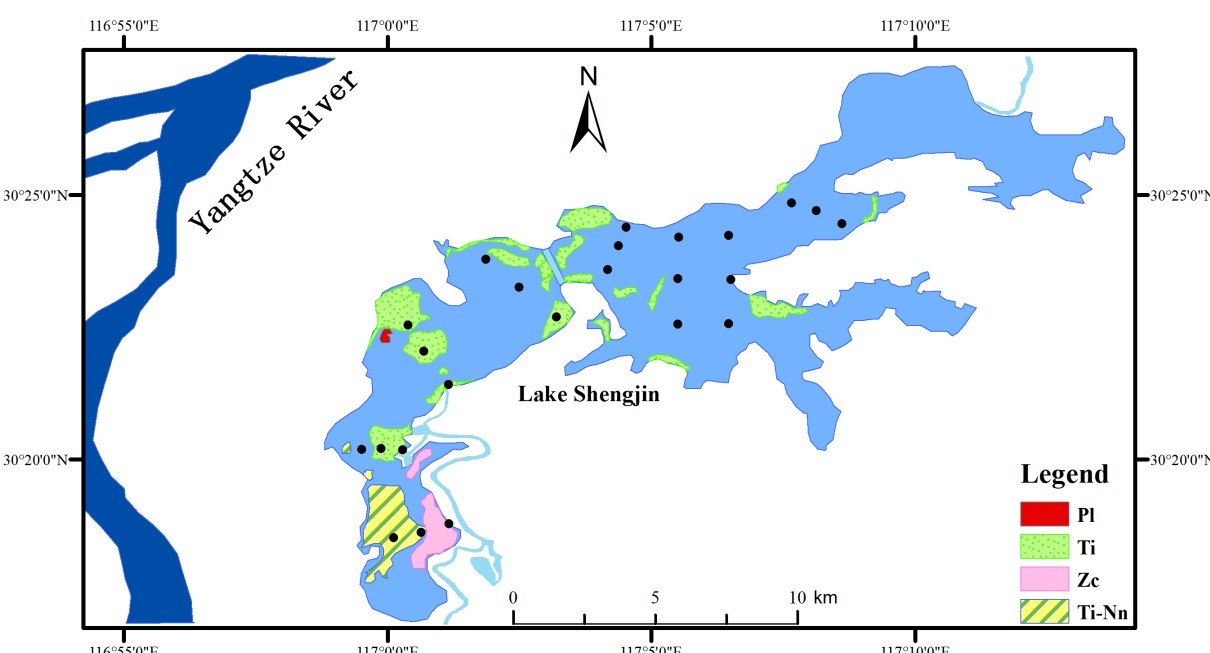

**Figure 2.** Distribution of dominant associations of main aquatic plants in summer. The following abbreviations used in the diagram are: Pl: *Polygonum lapathifolium*, Ti: *Trapa incisa*, Zc: *Zizania caduuciflora*, and Ti-Nn: *Trapa incisa-Nelumbo nucifera*.

### 3.3. Phytoplankton Community Diversity and Dominant Species

During the survey of this study (Figure 3), we identified a total of 254 species (including varieties) in 8 phyla and 137 genera of phytoplankton. The most numerous of these are Chlorophyta, with 109 species, accounting for 42.27% of the total species; this was followed by Bacillariophyta and Cyanophyta, with 56 and 34 species, accounting for 22.76% and 13.82%, respectively. In the summer survey, there were 174 phytoplankton species in the *T. incisa* range, 169 species in the *Z. caduuciflora* range, and 147 species in open water. Among them, we did not find *Dinobryon bavaricum*, *Pediastrum*, *Fragilaria capucina*, and *Melosira granulata* in most open waters, and we only detected *Actinastrum fluviatile* and *Strombomonas ensifera* in *Z. caduuciflora* distribution area.

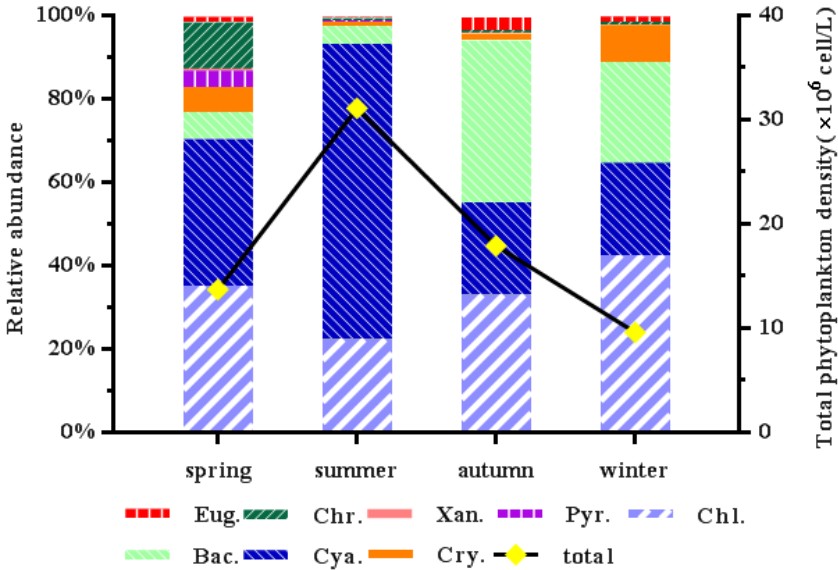

**Figure 3.** Total phytoplankton cell density and relative abundance ($\times 10^6$ cell/L) in Shengjin Lake.

The Shannon–Wiener index for phytoplankton in Shengjin Lake ranged from 1.45 to 3.52, with a mean value of 2.74. We recorded the high mean value, 2.96, in the winter. Spatially, the *T. incisa* range (3.06) > the *Z. caduuciflora* range (2.83) > the open water (2.41). The Pielou homogeneity index fluctuated from 0.55 to 0.87, with a mean value of 0.71. Temporally, the highest mean value was recorded in winter, at 0.73. Spatially, there was no significant trend. The Margalef species richness index ranged from 2.15 to 3.52, with a mean value of 3.15. Once again, we noted that, in the winter, the mean value was at its highest with 3.24. Spatially, the *T. incisa* distribution area was the highest, at 3.22. In summary, the results of the diversity indices illustrated that Shengjin Lake had the highest average in winter and the diversity index of the area with aquatic vegetation was higher than that of the open water area.

The analysis of the dominant phytoplankton species during the study period showed that they varied greatly among seasons (Table 2). In spring, the dominant species were *Merismopedia marssonii*, *P. pusillum*, and *D. bavaricum*. In summer, *Planktolyngbya subtilis*, *Phormidium tenuis*, and *Microcystis aeruginosa* dominated the Cyanophyta, *Selenastrum minutum*, and *Staurastrum indentatum* dominated the Chlorophyta. *P. subtilis* is the most dominant species, with the $Y = 0.22$, and the relative abundance of *P. subtilis* is more than 30% in the whole lake area with a detection frequency of 95.8% at each sampling. Aquatic vegetation grows vigorously in the summer, and there were significant differences in the distribution of aquatic vegetation in the lake area. Upon further analysis of the spatial distribution of the dominant species based on the results, we observed that there were spatial differences in the dominance index of *P. subtilis*, among which the *T. incisa* distribution area (0.24) > *Z. caduuciflora* distribution area (0.19) > open water area (0.11). In autumn, the dominant species shift from Cyanophyta to Bacillariophyta and Chlorophyta. This decreased the dominance of the *P. subtilis* ($Y = 0.03$), while *M. granulata* became the main dominant species with a relative abundance of 22%. *M. tenuissima*, *Synedra acus*, *M. granulata Her*, *Ankistrodesmus convolutus*, *Scenedesmus quadricauda*, *A. fluviatile* and *Cryptomonas ovata*, etc., were the dominant species in the winter. The dominant species differed among seasons, and we found the frequencies of the main dominant species to be similar in the UL and ML, but the dominant values of the dominant species varied among sampling sites.

**Table 2.** Dominant species and dominance of phytoplankton communities in each season.

| Species | | Dominance | | | |
|---|---|---|---|---|---|
| | | Spring (April) | Summer (July) | Autumn (October) | Winter (January) |
| Cyanophyta | *P. subtilis* | 0.012 | 0.228 | 0.032 | 0.019 |
| | *P. tenuis* | 0.003 | 0.035 | 0.014 | 0.01 |
| | *M. tenuissima* | 0.002 | 0.005 | 0.011 | 0.033 |
| | *M. marssonii* | 0.064 | 0.031 | 0.002 | 0.004 |
| | *M. aeruginosa* | - | 0.031 | - | 0.004 |
| Chlorophyta | *S. minutum* | 0.0007 | 0.035 | 0.006 | 0.017 |
| | *S. indentatum* | 0.005 | 0.025 | 0.009 | - |
| | *Chlorella vulgaris* | 0.015 | 0.011 | 0.059 | 0.019 |
| | *S. arcuatus* | 0.008 | 0.0006 | 0.035 | 0.002 |
| | *A. convolutus* | 0.006 | 0.017 | 0.012 | 0.043 |
| | *S. quadricauda* | 0.016 | - | 0.0031 | 0.081 |
| | *A. fluviatile* | 0.1 | 0.001 | - | 0.065 |
| Bacillariophyta | *M. granulata Her* | 0.002 | - | 0.134 | 0.034 |
| | *M. granulata* | 0.005 | 0.003 | 0.051 | 0.018 |
| | *S. acus* | 0.013 | 0.002 | 0.019 | 0.073 |
| Cryptophyta | *C. ovata* | 0.015 | 0.009 | 0.01 | 0.064 |
| Pyrrophyta | *P. pusillum* | 0.032 | 0.008 | - | - |
| Chrysophyta | *D. bavaricum* | 0.079 | 0.01 | 0.0002 | - |

### 3.4. Spatial and Temporal Variation of Phytoplankton Communities

The results showed (Figure 3) that there were seasonal differences ($p < 0.05$) in the mean cell density of phytoplankton at each sampling site, with the highest mean cell density of $3.11 \times 10^7$ cell/L in the summer, decreased gradually to $1.79 \times 10^7$ cell/L in the autumn, and the lowest of $0.96 \times 10^7$ cell/L in the winter. Cyanophyta was a major component of the phytoplankton community, accounting for more than 20% of the relative abundance in all seasons. The relative abundance of Cyanophyta in the summer was more than 70%, 47.72% of Cyanophyta are *P. subtilis*. The relative abundance of Cyanophyta gradually decreased in the autumn and reached the lowest value of 22.54% in the winter, and recovered in the spring. The relative abundance of Chlorophyta increased from 22.15% in the summer to the highest value of 42.57% in the winter. The relative abundance of Bacillariophyta climbed to the highest value of 38.29% in the autumn. *M. granulata* accounted for 48.84% of the abundance of Bacillariophyta. The number of Cryptophyta increased in the winter, and the relative abundance rose to 8.8%. Although Cyanophyta had absolute dominance in the summer, it did not in all the other seasons, allowing the relative abundance of Chrysophyta, Euglenophyta, Xanthophyta, and Pyrrophyta to increase.

The phytoplankton biomass of Shengjin Lake spanned from 0.78–15.52 mg/L with seasonal variation, reaching a maximum in the summer and then gradually decreasing, with similar trends in phytoplankton cell abundance.

In the following figure (Figure 4), the spatial distribution of the annual mean phytoplankton cell density in Shengjin Lake was significantly different ($p < 0.05$), with the annual mean phytoplankton cell density in the UL area being $1.76 \times 10^7$ cell/L. The trend shows that the mean cell density is higher in the central open water area, with a mean cell density of $2.02 \times 10^7$ cell/L, and lower in the surrounding aquatic vegetation cover area, with the lowest value found in the SJ1, where the water-holding plant experienced a good recovery, with a cell density of $1.51 \times 10^7$ cell/L. The phytoplankton cell density in the ML was generally higher than that in the UL, with an annual average cell density of $1.95 \times 10^7$ cell/L and the highest value of $2.24 \times 10^7$ cell/L, which was found in the SJ23, showing an obvious gradient distribution characteristic.

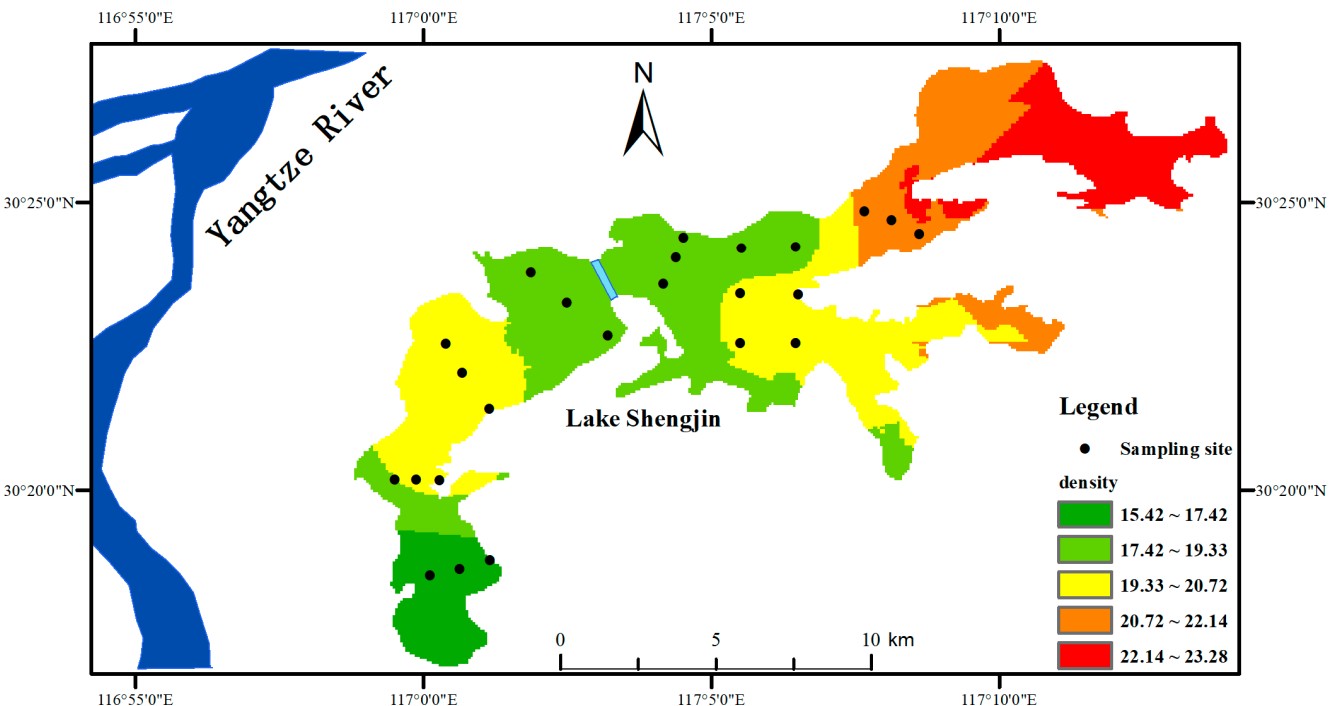

**Figure 4.** Distribution of annual mean phytoplankton cell density in Shengjin Lake ($\times 10^5$).

### 3.5. Relationship between Phytoplankton Communities and Environmental Parameters

In this study, we selected a total of 18 phytoplankton species (dominant species) and 14 environmental factors to analyze the relationship between the dominant phytoplankton species and environmental parameters throughout the year in Shengjin Lake [31,32]. We analyzed the relationship between the phytoplankton species, and the environmental parameters was analyzed using RDA. The adjusted variance explained 80.9%. Sort axes 1 and 2 explained the 22.9% and 31.5% of the variance in phytoplankton (Table 3).

**Table 3.** Results of RDA analysis of dominant phytoplankton species.

| Sort Axis | 1 | 2 | 3 | 4 |
|---|---|---|---|---|
| Eigenvalue | 0.329 | 0.211 | 0.126 | 0.101 |
| Species-environment correlation | 0.852 | 0.757 | 0.682 | 0.714 |
| Cumulative percentage of species | 22.9 | 31.5 | 36.3 | 38.6 |
| Cumulative percentage of species-environment relationships | 43.4 | 71.9 | 83.8 | 91.2 |

We conducted Monte Carlo tests conducted on phytoplankton communities and the environmental parameters, where WL, WT, TN, and Cond were the main parameters ($p < 0.05$). The results of the analysis showed that (Figure 5) the abundance of cells of the dominant species, *P. pusillum* and *D. bavaricum*, was significantly and positively correlated with NO$_3$-N; *P. subtilis* was positively correlated with *T. incisa*, SD and WT; *S. indentatum* and *P. tenuis* were positively correlated with *Z. caduuciflora*, WT and WL; *M. granulata* and *S. acus* were positively correlated with TN, DO, and the significant negative correlations were found between SD and *T. incisa*.

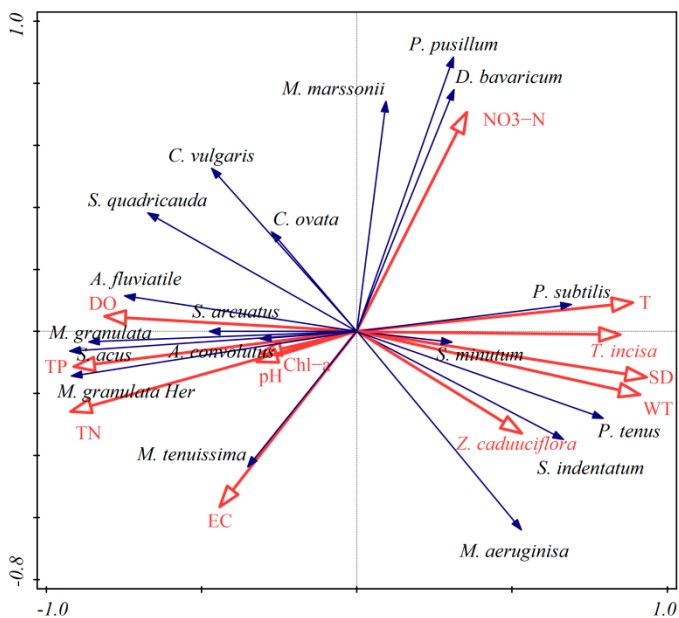

**Figure 5.** A sequencing map of the RDA phytoplankton species (blue lines with arrowhead) and the environmental parameters (red lines with arrowhead).

## 4. Discussion

### 4.1. Aquatic Vegetation Restored after Removal of Purse Seine

Since the removal of the seine net in 2018, the aquatic vegetation of the whole lake surface increased significantly. Compared to 2015, the area of aquatic vegetation increased to 26 km$^2$, and the coverage of aquatic vegetation was more than 50%, especially the coverage of the emerged plant *Z. caduuciflora* in the UL waters of sampling points SJ1~3 in the UL was more than 90%. The purse seine fishery, which began in 1997, led to severe

degradation of aquatic vegetation. In the 2007 monitoring survey, the area of purse seine farming had reached 100%, and the aquatic vegetation cover was only 30%. There were large areas of water that were vegetation-free in the UL, and only the channel in the ML contained small amounts of the floating plant growth [33,34]. With the advent of the fishing ban policy in 2015 in the UL, the submerged plants *C. demersum* and *M. verticillatum* have begun to appear in patches, the distribution area has expanded from the central to the northern part, and we found only a small amount of the emerged plant Z. *caduuciflora* between the purse seine in the ML [35]. According to the data, Hangzhou West Lake and Wuhan East Lake also experienced serious damage to aquatic vegetation caused by the over-farming of fish and later restored aquatic vegetation by reducing fish density.

According to the available data, the early stage of the commercial use of Shengjin Lake was Chinese velvet chelonia farming, and the middle and late stages developed mainly into fishery seine farming. With the bridge as the boundary, seine nets were deployed only in the junction area of the UL and the ML. Extensive contract fishery culture was implemented in the UL, while numerous artificial dikes and culture nets were constructed in the ML. The lake was divided into multiple culture areas with intensive seine nets for high-density aquaculture throughout the year. Filter-feeding fish such as *Hypophthalmichthys molitrix* and *Aristichthys nobilis* were the main dominant species [36]. The small number of seine nets in the UL and smaller catches than in the ML may be the reason for the presence of aquatic plants. After the removal of seine nets in Shengjin Lake in 2018, the density of fish decreased, the feeding of herbivorous fish on aquatic plants decreased, and aquatic plants were able to grow and renew. At the same time, after the ban on aquaculture, pollution from bait placement was alleviated to a greater extent, the lake water quality improved, aquatic plants adapted to water quality conditions, and aquatic vegetation recovered rapidly, especially submerged vegetation.

*4.2. Phytoplankton Community Composition and Spatial and Temporal Variation*

The findings of the current study revealed large seasonal differences in phytoplankton communities, and similar variations have been identified in previous surveys [37,38]. Compared to the 2014 survey data, the number of phytoplankton species increased from 210 to 254, with an increase in the number of species of the Chlorophyta and Euglenophyta. The decrease in phytoplankton cell abundance and the increase in biodiversity was due to the improvement in water quality after the removal of the purse seine ban.

According to the data of previous surveys [39], the TP concentration in Shengjin Lake decreased by 20% (0.053 mg/L to 0.41 mg/L). The average transparency increased from 0.4 m to 0.7 m, and some waters covered by aquatic plants can reach 1.2 m. Phytoplankton is highly sensitive to environmental changes. The changes in environmental factors after the removal of purse seine affect the structural distribution of the phytoplankton community.

Compared with before and after the removal of the purse seine, the species of Chlorophyta are the most abundant, with the highest Cyanophyta cell density after purse seine removal in summer. Shengjin Lake has a high water level in the wet season in summer, and the water temperature exceeds 35 °C, which is most suitable for the growth of Cyanophyta. At the same time, after the purse seine is removed, the coverage area of aquatic vegetation is greatly increased, which has enhanced the purification of lake water quality, high water transparency, and strong photosynthesis of Cyanophyta. The main dominant species in the summer after the removal of the purse seine is *P. subtilis*, which is significantly less than that of other cyanobacteria before the removal of the purse seine. *P. subtilis* belongs to *Lyngbya*, and there is a gelatinous sheath inside the cells that regulate the intensity of light received. This may be the reason why *P. subtilis* is the dominant species in the summer [40].

As the number of aquatic plants increased significantly after the removal of the seine, almost all aquatic plants died in autumn and winter, and the nitrogen and phosphorus from the plant body entered the water body, resulting in higher nitrogen and phosphorus concentrations compared to the seine period. After the removal of the purse seine, the *M. granulata* replaced Chlorophyta as the dominant species in autumn due to the appro-

priate temperature and high nutrient level. The RDA results showed that Bacillariophyta cell abundance correlated negatively with WT and SD. When the water of Shengjin Lake flowed into Yangtze River, the WL in most of the lake area was at 1 m, the WT was below 15 °C, the aquatic vegetation disappeared, and the water environment favored favorable for the growth of Bacillariophyta [41]. Compared with before the removal of the purse seine, the number of phytoplankton species increased, and *Scenedesmus* became the dominant species in winter, probably because *Scenedesmus* thrives in a low-temperature environment, the chemosensory effect of *Scenedesmus* decreased with the disappearance of aquatic plants [42,43].

### 4.3. Response Mechanisms of Phytoplankton Communities to Aquatic Vegetation Restoration

Studies have shown that aquatic vegetation indirectly affects the community structure of phytoplankton by altering the lake water environment. The restoration of floating and submerged aquatic vegetation in Shengjin Lake provides diverse habitats for the growth and development of phytoplankton, resulting in increased phytoplankton diversity. The results of RDA analysis that the content of nitrogen and phosphorus correlated positively with the growth of *M. granulata* and most Chlorophyta, and the nutrient concentration directly affected the phytoplankton community structure. The death of aquatic plants in the autumn and winter to produce organic matter explained the high cell abundance of Bacillariophyta and Chlorophyta during those seasons, while the fixation of suspended matter on the lake surface by aquatic vegetation disappeared, and wind and rainfall caused the release of nitrogen and phosphorus from the bottom mud [44]. The cell abundance of the dominant species of *P. tenuis* and *P. subtilis*, was negatively correlated with DO and positively correlated with the floating plant *T. incisa*. Several sites covered with *T. incisa* in the summer identified that the relative abundance of the algae *P. subtilis* was higher than at the open water sites. Aquatic plants consume a large amount of DO and reduce pH when decomposing organic matter and nutrients [45]. As a result, the available light of phytoplankton in the area with high *T. incisa* coverage in the UL is less, the respiration rate of phytoplankton is greater than that of light cooperation and the photosynthesis and respiration of aquatic plants and phytoplankton jointly affect the trend of DO.

Aquatic plants also directly influence the community structure of phytoplankton through themselves. In the summer, we identified a high abundance of *D. bavaricum* and Pyrrophyta in the southern sampling sites covered by *T. incisa* and *Z. caduuciflora* in the UL. The fixation type algae such as *P. subtilis* and *Dinobryon* have flagella or fixation filaments. Aquatic plants provide a habitat for the large fixation type phytoplankton, and algae attach to aquatic plants to avoid predation by fish [46]. The phytoplankton identified in the distribution waters of aquatic vegetation, such as *Pediastrum* and Euglenophyta, are mostly above 10 μm in diameter, have a fresh weight of more than $0.01 \times 10^{-6}$ mg, and are presumably shade and cold-tolerant species.

Aquatic plants improve water quality and also compete with phytoplankton for light and nutrients. The three UL sampling sites (SJ1~3) with high aquatic vegetation cover in the summer have low phytoplankton abundance, with an average cell density of $1.47 \times 10^7$ cell/L. Aquatic plants provide a sheltered environment for zooplankton and filter-feeding fish, causing smaller phytoplankton to be more easily predated and the abundance of larger Chrysophyta and Pyrrophyta to increase relatively. Some studies have shown that filter-feeding fish tend to make phytoplankton larger under low nutrient conditions. On the other hand, aquatic plants and phytoplankton have similar ecological niches, and both may secrete chemosensitive substances to inhibit each other's growth [47]. There are other studies detailing how the main components of chemosensitive substances in aquatic plants, such as *N. nucifera* and *C. demersum*, include secondary metabolites such as flavonoids and organic acids [48,49], which can damage the cell structure of Cyanophyta and other phytoplankton, affect intracellular enzyme activities, and prevent their photosynthesis [50,51].

## 5. Conclusions

The results showed that the spatial and temporal changes in the phytoplankton community structure of river-connected lake were in response to the restoration of aquatic vegetation and changes in environmental factors in the lake. After the seine was removed, the aquatic plants re-grew. The direct and indirect effects of aquatic plants are one of the important causes of changes in the structure of phytoplankton communities. Monitoring phytoplankton to evaluate the restoration of the middle and lower reaches of Yangtze River through-river lake ecosystem is an important tool in the current lake ecological restoration project, and conducting research related to aquatic plants and phytoplankton has important ecological significance and value.

**Author Contributions:** Conceptualization: Z.Z. and X.Y.; data curation: X.Z., J.C. and W.G.; investigation: X.Z., J.C. and W.G.; methodology: Z.Z.; writing—original draft: X.Z.; writing—review and editing: X.Z., Z.Z. and X.Y.; funding acquisition: Z.Z. All authors have read and agreed to the published version of the manuscript.

**Funding:** This research was supported by Joint Research Project for the Yangtze River Conservation (Phase I), China (No. 2019-LHYJ-01-0212, 2019-LHYJ-01-0212-17).

**Institutional Review Board Statement:** Not applicable.

**Data Availability Statement:** The data presented in this study are available on request from the corresponding author Xiaoxin Ye. E-mail: yexx@ahu.edu.cn.

**Acknowledgments:** We would like to thank Jingwen Chen, Wenli Guo for their assistance with data collection. We would also like to thank Zhongze Zhou and Xiaoxin Ye for their advice on the paper. We would like to thank Marci Baun from the University of California, Los Angeles, for editing the paper.

**Conflicts of Interest:** The authors declare no conflict of interest.

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
