# Peer review of "Response of Phytoplankton Community Structure to Vegetation Restoration after Removal of Purse Seine in Shengjin Lake"

_diversity, doi:10.3390/d14030178_

Round 1

Reviewer 1 Report

See attached file

Author Response

Thank you very much for your comments on my paper, which has been of great help to me in revising my paper. I checked the statistical analysis methods with references and laboratory data to determine the validity of the statistical analysis in the paper, and explained the findings in the paper. I have revised my paper according to your comments, and the results are as follows:

MAJOR POINTS

L162: All terms in the equation have been explained. The order of operations has been explained.  0.02 is the phytoplankton dominance index (Y). References have been provided.

fi is the frequency of species.This is used to calculate the phytoplankton dominance index (Y).

L162-167: The predictor variables are environmental parameters of each season. The response variables are phytoplankton communities cell density of each season.

L168: Pearson correlation analysis is used to test the relationship between water quality parameters and phytoplankton density. It has been briefly illustrated in the paper.

L172-173: WINDOWS program refers to Canoco for Windows 5 software, i.e. Canoco 5. Modifications have been made in the paper.

L185-187: Thank you for your comments on this point, I have revised and explained this point in the paper. We used one-way ANOVA to analyze the water quality parameter data of each sampling point in each season. The analysis results showed that the data of WL and SD in summer were significantly higher than those in other seasons. Compared with other seasons, the p value was less than 0.05, indicating that there was a significant difference at the level of 0.05. At the same time, Pearson analysis testd that the result was * P < 0.05, indicating a significant correlation at the level of 0.05 (bilateral). The average value of water quality parameters in Table 1 is convenient to visually see the data changes between seasons.

In Fig.2: For this separation phenomenon in the Fig.2, the figure shows the distribution of main aquatic plant associations,there are other associated species, but their coverage and density are far lower than the dominant aquatic plant associations,their influence on water quality parameters is not as great as that of dominant species, so they are not shown in the figure, which has been explained in result 3.2. For the relationship between aquatic vegetation distribution and niches, I explained that 1) The dominant aquatic plants are emergent plant Zizania caduuciflora and submerged plant Trapa incisa. The water depth in the upper Lake area is low, especially in the south of the upper Lake area, which is only 3.7 ~ 4 meters. The area with high coverage of aquatic vegetation is close to the shore, which is suitable for the growth of emergent plants. 2)The submerged plant Trapa incisa and emergent plant Zizania caduuciflora were also highly distributed in Shengjin Lake before the removal of the purse seine. After the removal of the purse seine, the seeds and fruits of Trapa incisa and Zizania caduuciflora at the bottom of the lake grew first, became a pioneer species and better adapted to the water environment, so Trapa incisa and Zizania caduuciflora became the dominant species.

L220-221: This is explained as a phenomenon caused by biological succession. Zizania caduuciflora and Trapa incisa belong to thermophilic plants. They can grow at 10-25 ℃ and are not resistant to cold, so they will wither in autumn and winter. The growth period of Potamogeton crispus is different from that of most aquatic plants,Potamogeton crispus is a typical submerged plant that germinates in autumn and grows through winter. Potamogeton crispus mainly depends on budding, which germinated in early October. I have provided biological insights in the results.

L304-305: The phytoplankton selected in RDA analysis are species with dominance index greater than 0.02. They were identified at most sampling sites. After years of experiments by our research team, this analysis method is suitable for the study of phytoplankton community in freshwater lakes. I have provided some references.

Thank you very much for your references, which is very useful for the analysis of the results of my paper. I will study and revise these references and apply them to future papers.

MINOR POINTS

L18: I have modified the sentence format.

L31: Abbreviations have been expanded.

L46: I have deleted redundant instance.

L88: I have expanded the abbreviations.

L181: I have inserted a space.

L249: I have inserted a space.

L 257-258: I have checked and revised this sentence.

L259: I have replaced the word.

L272-273: I have replaced this sentence.

L308-309: First two sort axes(axes 1 and 2) in Table 3 represent the quadrant axes of the coordinates in Fig.5, which are used to analyze the correlation between environmental factors and sorting axes. They are the linear combination of environmental factors generated by Canoco5 software.

Reviewer 2 Report

I appreciate the authors for taking a environmental impact study and presenting it very well. The manuscript was well planned and designed. I have put my comments in the pdf and requesting you to address the same.

Author Response

Thank you very much for your comments on my paper, which has been of great help to me in revising my paper. I have revised my paper according to your comments, and the results are as follows:

L31: I have expanded the abbreviations.

L41: I have modified the word.

L46: I have deleted Extra words.

L47: I have revised the sentence.

L50、88: I have expanded the abbreviations.

L140~143: I have expanded the abbreviations.

L203: April is the month when the rainfall of Shengjin Lake has just begun to increase. However, the maximum rainfall of Shengjin Lake occurs in July and August in summer, And aquatic plants grow vigorously. The previous sentence has been clarified.

L208: The word has been changed to full form.

L231: I have deleted Extra words.

L336: I have changed the italics.

L370: I have modified repeated content.

Round 2

Reviewer 1 Report

Reviewer's comments for "Response of phytoplankton community structure to vegetation restoration after removal of purse seine in Shengjin Lake" by and co-workers

General appraisal

In the revised version, the authors have addressed most of my concerns with the previous version of the manuscript, but I still have a few questions that require clarification. I also suggest that the manuscript be checked for English should it make it through the review process.

Specific comments

[1] L165: P_i (with upper-case P) does not appear the equation. Is it a typo? (maybe it should be p_i with lower-case p). 

[2] Is Y=p_i*f_i equivalent to the formula (1)?

[3] In formula (1), should Y on the left-hand-side be Y_i? (I think that the index i is missing here)

[4] L167: Do you mean "following Lampit et. al (1993), we consider Y>=as an indication of dominance?

[5] Following from the above comment, the choice of 0.02 as threshold for dominance may be applicable to a particular community. I wonder whether it is valid for any community. I suggest that the authors check this out.

Author Response

Response letter

Dear Editors and Reviewers:

Thank you for your comments concerning our manuscript entitled ‘Response of phytoplankton community structure to vegetation restoration after removal of purse seine in Shengjin Lake’. These comments are all valuable and very helpful for revising and improving our paper, as well as the important guiding significance to our researches. We have studied comments carefully and have made correction which we hope meet with approval. We have checked the English language in manuscript and have polished the paper in English through Marci Baun from University of California, Los Angeles, and the results are as follows:

Thanks again!

Sincerely,

Xudong Zheng

Corresponding author: Xiaoxin Ye

[1] L165:Pi is a typo. I have modified Equation 2.

[2]  Equivalent. I have modified Equation 2.

[3] Thank you for your reminder. I have searched relevant information and articles, and I have confirmed that Y does not need to add index i.

[4] L167: I have revised the sentence.

[5] Regarding your question, I checked Y for Mcnaughton dominance index by reviewing relevant literature and confirmed that the dominant species criterion for phytoplankton is Y > 0.02, and this threshold applies to all phytoplankton species.

Thank you again for your positive and constructive comments and suggestions on our manuscript. we hope you will find our revised manuscript acceptable for publication.